# FAPNET: Feature Fusion with Adaptive Patch for Flood-Water Detection and Monitoring [note 1]

**DOI:** 10.3390/s22218245

**Published:** 2022-10-27

**Authors:** MD Samiul Islam, Xinyao Sun, Zheng Wang, Irene Cheng

**Affiliations:** 1Multimedia Research Centre, University of Alberta, Edmonton, AB T6G 2E8, Canada; 23vGeomatics Inc., Vancouver, BC V5Y 0M6, Canada

**Keywords:** flood-water mapping, waterbody detection, satellite image analysis, image segmentation, SAR imagery

## Abstract

In satellite remote sensing applications, waterbody segmentation plays an essential role in mapping and monitoring the dynamics of surface water. Satellite image segmentation—examining a relevant sensor data spectrum and identifying the regions of interests to obtain improved performance—is a fundamental step in satellite data analytics. Satellite image segmentation is challenging for a number of reasons, which include cloud interference, inadequate label data, low lighting and the presence of terrain. In recent years, Convolutional Neural Networks (CNNs), combined with (satellite captured) multispectral image segmentation techniques, have led to promising advances in related research. However, ensuring sufficient image resolution, maintaining class balance to achieve prediction quality and reducing the computational overhead of the deep neural architecture are still open to research due to the sophisticated CNN hierarchical architectures. To address these issues, we propose a number of methods: a multi-channel Data-Fusion Module (DFM), Neural Adaptive Patch (NAP) augmentation algorithm and re-weight class balancing (implemented in our PHR-CB experimental setup). We integrated these techniques into our novel Fusion Adaptive Patch Network (FAPNET). Our dataset is the Sentinel-1 SAR microwave signal, used in the Microsoft Artificial Intelligence for Earth competition, so that we can compare our results with the top scores in the competition. In order to validate our approach, we designed four experimental setups and in each setup, we compared our results with the popular image segmentation models UNET, VNET, DNCNN, UNET++, U2NET, ATTUNET, FPN and LINKNET. The comparisons demonstrate that our PHR-CB setup, with class balance, generates the best performance for all models in general and our FAPNET approach outperforms relative works. FAPNET successfully detected the salient features from the satellite images. FAPNET with a MeanIoU score of 87.06% outperforms the state-of-the-art UNET, which has a score of 79.54%. In addition, FAPNET has a shorter training time than other models, comparable to that of UNET (6.77 min for 5 epochs). Qualitative analysis also reveals that our FAPNET model successfully distinguishes micro waterbodies better than existing models. FAPNET is more robust to low lighting, cloud and weather fluctuations and can also be used in RGB images. Our proposed method is lightweight, computationally inexpensive, robust and simple to deploy in industrial applications. Our research findings show that flood-water mapping is more accurate when using SAR signals than RGB images. Our FAPNET architecture, having less parameters than UNET, can distinguish micro waterbodies accurately with shorter training time.

## 1. Introduction

Flooding often results in devastating consequences. Although societies have learnt over time how to create flood-prevention infrastructure, many areas continue to be affected by these disasters, costing property damage and lives. Climate change-induced flooding, whether from rising sea levels, extreme weather events, or shifting tides, is likely to affect nearly 4.8 million Canadians who live within 10 kilometers of the eastern or western coastline in the near future, according to a report published by the University of Waterloo [1]. Not only in Canada but also all around the globe, flooding is becoming increasingly common and can be devastating. Promisingly, studies show that it is possible to predict how severe the flooding may be based on the information available [2]. A combination of computer vision and machine learning techniques can be used to predict the occurrence of flooding, so that residents can evacuate [3].

Water level change and flood detection have traditionally been analyzed using imagery from passive sensors, such as optical images, applying approaches like the Normalized Difference Water Index (NDWI) and other water indices [4]. There are a variety of ways to combine multiple frequency bands to create signatures that are sensitive in detecting waterbodies [5]. Once an index has been calculated, it is possible to apply thresholds to detect water changes and flooding throughout a region. However, their use is confined to a specific time period and specific weather conditions, making them less ideal for rapid reaction in dynamic scenarios. As a result, Synthetic Aperture Radar (SAR) imagery has become a popular satellite data modality used for flood-water detection [6]. Because of its active sensing capability and specific frequency ranges, SAR images can be acquired at any time of the day despite cloud cover. As long as the image features can be extracted, high-resolution imaging is not always necessary.

Nevertheless, radar images are known to be impacted by polarized waves in the signal [7]. Our approach is to integrate the results of several polarization methods in order to provide a complete understanding of the ground surface, while a natural image offers three channels (red, green and blue). Polarization offers four combinations to analyze the separate channels: Vertical Transmission and Vertical Reception (VV), Vertical Transmission and Horizontal Reception (VH), Horizontal Transmission and Vertical Reception (HV) and Horizontal Transmission and Horizontal Reception (HH), where each channel provides unique characteristics. We may use one of these signals to detect flooding on individual chips but combining all the signals in semantic modeling will obtain accurate waterbody detection for flood mapping [8]. Supplementary elevation data (NASADEM) is a significant feature leading to higher accuracy for deep learning model training. Microwave signals are not suitable in conjunction with basic computer vision techniques such as indexing and thresholding.

In addition, GeoTiff radar signal preprocessing techniques have advanced rapidly in recent years. Radar imaging systems are often installed on aeroplanes or satellites with side-looking geometry. In radar imaging, shadow is a common geometric distortion that occurs in mountainous environments with high topographic variation. With the knowledge of topographic variation, it is feasible to extract waterbodies from shadows since the Digital Elevation Model (DEM) elevation values of most waterbodies tend to be the same [9]. For this reason, DEM was obtained from the corresponding coordinates of each polarized signal. Since microwave signals have a wide range of bands described as hyperspectral channels, data preprocessing becomes challenging and introduces additio rhead [10]. Deep learning models cannot directly manage a large number of channels. Hence, in this paper, we demonstrate that this issue can be addressed by combining (1) semantic modeling for selecting relevant channels (bands) and (2) efficient model design.

To obtain a high-level understanding of an image, semantic segmentation assigns each image pixel with a class label, which is known as pixel-wise classification [11]. Predefined features are frequently used in traditional machine learning applications for pixel-wise categorization [12]. A number of network architectures, including DenseNet [13], ResNet [14], EfficientNet [15] and VGG [16], have been introduced. These models are trained on different large datasets (e.g., imagenet [17]), which can be used as the encoder backbone in an image segmentation architecture to yield high segmentation accuracy. In recent years, there has been an increasing use of Deep Convolutional Neural Network (DCNN) models for waterbody identification [18,19,20,21,22]. According to the results and discussion [23], DCNN-based models are capable of extracting distinct spatial feature representations better than water index-based computer vision methods. In particular, FSSS has been shown to be the most effective technique for identifying water surfaces using satellite imagery or SAR.

Large intraclass differences and lower sensor resolution make it more difficult to identify objects in satellite images using image segmentation methods [24]. Recent advances include multimodality [25], hierarchical feature representations [26] and fusion techniques [25,27,28]. Three Convolutional Neural Network models are combined using MonteCarlo dropout uncertainty maps [29] to go beyond traditional weight averaging for land-cover mapping segmentation in urban zones. Kampffmeyer et al. [29] used a digital surface model established by Sun and Wang [30] to enhance the Fully Connected Network (FCN) to generate segmentation outcomes of Very-High-Resolution (VHR) remote sensed images.

A recent approach for flood-water mapping was proposed by Jiang et al. [31]. The authors used Sentinel-1 imagery, building their technique upon Attention-UNet3+ and using the Poyang Lake Region as a case study. They combined the feature maps from different scales to improve the accuracy of narrow waterbody detection. The authors claimed that by using a spatial attention module, false alarms caused by speckle noise and water shadows are reduced. However, the model requires a large number of parameters, which creates high computational overhead, with expensive GPU consumption during training. Instead, for narrow and tiny surface water detection, we use the NAP augmentation module and DFM during data preprocessing. We are able to obtain better results by adding external features provided by the Microsoft Planetary Data and DEM to reduce the number of shadow signals from the radar imagery. Another approach to detecting flood-water from RGB images was proposed by Sarp et al. [32], who used Mask-R-CNN. However, the authors only focused on detecting flood-water from roadways, which is for vehicle routing control and traffic management, in the smart autonomous vehicle application domain. To train their model, the authors used hand-crafted items. It is worth pointing out that their dataset is biased toward one class (background). Their evaluation was based on the F1 score, which is not guaranteed to be consistent with the MeanIoU score. Based on our study and the work of other researchers in the literature, MeanIoU is a reliable metric for segmentation performance evaluation because it calculates the average of each detected pixel, providing equal importance to each class. For this reason, we use the MeanIoU score for performance evaluation and comparison in our work.

In radar backscatter, there are similarities between flooded and unflooded conditions. Therefore, additional features are needed to boost the performance of the waterbody detection accuracy. Zhao et al. proposed Exclusion map (Ex-map), derived from Sentinel-I time series information, which aims to delineate the areas, where radar scatters do not detect flood-water [33]. By introducing Ex-map, the classification accuracy is increased significantly. Nevertheless, the lack of training data adversely affected the detection accuracy. To address this problem, a transfer learning-based flood detection technique, using RGB images, was proposed by Sazara et al. [34]. The authors suggested feature extraction techniques such as Local Binary Patterns (LBP), Histogram of Oriented Gradients (HOG) and pre-trained deep neural network (VGG-16), followed by water surface classification. Trained logistic regression, k-nearest neighbours and decision tree classifiers were used. However, optical RGB images contain less features than an SAR signal. SAR provides more information capable of wide area monitoring even at night as well as in worse weather conditions. Due to a larger spectrum of data, SAR signals require special preprocessing steps in order to benefit from the complex SAR features and achieve more accurate flood-water detection.

Considering the drawbacks of the current methods and based on our analysis, we propose a novel approach for a flood-water detection and monitoring system that embeds multi-channel fusion, unit kernel-based patch creation and a FAPNET model. Our method includes data fusion to handle multiple image resolutions and normalization of Sentinel-1 VV, VH and elevation data from the NASADEM bands. We also created a new NAP augmentation module to keep the high-dimensional representations prominent. NAP augmentation reduces patch generation data loss while improving model performance. By comparing with related methods [8], our experiments have shown better performance than existing deep learning approaches.

One novel idea introduced in this paper is the successful design and validation of FAPNET, which integrates the NAP augmentation algorithm and DFM. FAPNET can process both radar signals and optical images efficiently. However, the focus of this paper is on satellite SAR data analyzing invisible hyperspectral bands which provide more feature information. Applying FAPNET on the visible RGB bands for other applications can be studied in future work. NAP is a new and unique concept, which is different from the traditional augmentation method. It optimizes memory usage, which is important for industrial applications, where computing resources can be a bottleneck. We validated the better performance of our proposed model FAPNET by comparing it with related works, using qualitative, quantitative and memory usage analysis. Our contributions are summarized below;

In order to maximize the effectiveness of the learned weights, we introduce a multi-channel DFM using VV, VH and elevation data from the NASA Digital Elevation Model (NASADEM), incorporating Feature Fusion, normalization and end-to-end masking.Inspired by the CNN architectures, we introduce an effective augmentation technique by creating an NAP augmentation module to extract multi-receptive features as patches using unit kernel convolution at multi-scale levels, which helps to achieve an enriched semantic representation and helps the model to learn faster.We confirm the efficacy of our method by designing four experimental setups: CFR, CFR-CB, PHR and PHR-CB, which build a progressive analytic framework for researchers to investigate semantic segmentation in SAR images and conduct comparisons with state-of-the-art methods.We propose FAPNET, which is a lightweight, memory-time efficient and high performance model, as demonstrated in our quantitative and qualitative analysis.

The rest of this article is structured as follows: Section 2 explains data preparation, feature mapping and data augmentation. Details of our proposed Fusion Adaptive Patch Network (FAPNET) model and NAP augmentation algorithm are explained in Section 3. Section 4 provides quantitative, qualitative and memory-time performance assessments and Section 5 discusses our findings. Conclusion and future work are given in Section 6.

## 2. Study Area and Dataset

We used the “Map flood-water from Radar Imagery” competition dataset, which is hosted by Microsoft AI for Earth [8]. The dataset consists of Sentinel-1 images, masks and a CSV file containing metadata such as city and year. Sentinel-1 images and masks were collected from different regions of the world between 2016 and 2020. Sentinel-1 is able to send and receive signals in both horizontal and vertical polarizations since it is a phase-preserving dual polarisation SAR device. The dataset contains 542 chips as features and corresponding flood-water mapping as masks. Based on how radar microwave frequencies are dispatched and received, a single chip characterizes two bands or images (VV, VH). Each image is saved as a 512×512 resolution GeoTIFF file and each image pixel value denotes the amount of energy, expressed in decibels, reflected back to the satellite from the ground. Chips from 13 flood occurrences are included in the dataset and each flood event contains about 15 to 69 chips (30 to 138 images), with half of the events comprised of fewer than 32 chips (64 images). Table 1 shows a summary of the dataset divided by locations.

### 2.1. Feature Mapping and Discrepancy in Masking

According to our observations in Figure 1, it is obvious that the dataset has some issues, i.e., surface water information is missing in the features (VV, VH, DEM) but available in the corresponding masks. When visualizing features with their respective masks, it is noticeable that the extracted features indicate fewer water pixel values than the mask. Figure 1 illustrates a few examples. Figure 1c is an example of a consistent feature map and mask from the dataset. However, in Figure 1a,b, we can see discrepancies between the features and masks; in particular, Figure 1a shows significant differences. Moreover, several feature values in Figure 1b do not correspond to the water information correctly and the right-hand-side white patch of pixels in the VV and VH images reflects missing values, which might be due to human error or distortion caused by misinterpreted radar frequency.

To solve this problem, a perfect mask (label) creation is required, and we need a unique technique to minimize the errors. Consequently, we introduce a novel data augmentation technique called the NAP augmentation algorithm, which will be discussed briefly in Section 3.2. Note that the NAP augmentation algorithm is designed not only to minimize the error of mislabeling but also to boost the performance. In addition, many surface characteristics, such as elevation, terrain and ground property, can cause water to disperse or accumulate in one place; we tried several techniques and different experimental setups to evaluate the model performance. Our finding shows that, given the different water level changes in various regions, accurate feature mapping dictates the success of water pixel or flood prediction.

### 2.2. Data Augmentation

Albumentations is a fast and flexible data augmentation tool commonly used in industrial projects, especially in open source projects. Most image transform operations and certain domain or task-specific analyses, such as weather change predictions for autonomous vehicle perception modeling, are supported by Albumentations. Despite other libraries applying bilinear interpolation, its resizing and rotating process can adversely affect the task outcome because such data augmentation approaches might modify the mask value of the associated augmented feature [35]. Buslaev et al. [35], on the other hand, suggest employing regression unit tests to validate the augmented mask. Moreover, they reported state-of-the-art results for the segmentation task on the Inria Aerial Image Labeling dataset [36] after using several augmentation methods. A regression unit test is included in the Albumentations tool. We will describe our methodology in the next Section.

## 3. Methodology

In Section 2, we explained the study area and other important information regarding the dataset. The Microsoft AI for Earth Competition identifies VV, VH and DEM as important features for flood-water detection and therefore we use the same input features in our experiments [8]. Competition also highlights the significance of both signals and the need for DEM data in order to achieve improved accuracy, particularly when training deep learning models. Natural terrain and persistent water sources in a region’s environment might further aid a deep learning model’s ability to identify flooding. When processing the raw images, we included elevation information from NASA’s Digital Elevation Model (NASADEM). As illustrated in Figure 2, data is passed through the Albumentations tool [35] for augmentation and is further split into training (80%), validation (10%) and testing (10%) samples. In order to measure the model’s competence while tuning its hyperparameters, we used a validation dataset, which was a subset of the data used during training. In order to objectively evaluate how well a model fits the training data, a separate test dataset was deployed. In this study, we compared the outcomes based on the test dataset to provide a comprehensive evaluation.

### 3.1. Overall Strategy

In this work, we introduce the FAPNET model, which comprises two new techniques: the NAP augmentation method and the DFM. We fuse three single channel signals, VV, VH and DEM, in the DFM. Then we normalize the Feature Fusion as shown in Figure 2. In order to evenly distribute the classes in the dataset, we swap the unlabeled pixel values of the mask into the background. Next, we apply the NAP augmentation algorithm, which will be explained in Section 3.2. In order to evaluate our proposed FAPNET model, we designed four distinct experimental setups CFR, CFR-CB, PHR and PHR-CB and tested across a wide range of water dynamic scenarios. We compared FAPNET with existing segmentation models—UNET, VNET, DNCNN, UNET++, U2NET, ATTUNET, FPN and LINKNET. CFR and CFR-CB take only 512 × 512 input before running the DFM module. PHR and PHR-CB take 256 × 256 input before running the DFM and NAP augmentation algorithm. The same parameter settings and preprocessing techniques are used in all experiments. The four distinct experimental setups serve to validate and compare different models’ performances in various real-world contexts. The assessment metrics are MeanIoU and focal loss, which will be further discussed in the Quantitative Assessment section. Our FAPNET model is inspired by the encoder−decoder-based structure of UNET, but the number of hidden layers and convolution layers is different; FAPNET analyzes more deeply into the hyperparameter settings as described in Section 3.4. We validated our FAPNET model performance using quantitative, qualitative and memory-time efficiency analyses.

### 3.2. Proposed NAP Augmentation Algorithm

The majority of the augmentation techniques use a bilinear interpolation method that can affect data analysis and evaluation outcomes. To address this issue, Albumentations [35] recommends a regression unit test for validation. However, this does not guarantee the correct transformation of pixel values when arbitrarily rotating or flipping images. Inspired by the CNN architecture [37], our proposed NAP augmentation module makes use of a unit kernel and pixel-wise convolution that does not transform the pixel values and instead extracts the pixel values as patch images.

We use the NAP augmentation module to extract patches of defined size from each image in the dataset. Current patch libraries that we have explored so far lack the capability to overlap extracted features and do not ensure class balance. Some tools distort feature values while extracting image patches. In our NAP augmentation module, we deploy the CNN architecture concept and preserve additional parameters that make it possible to extract overlapping features and maintain class balance.

Note that any factor like the input shape, the kernel, zero padding, dilation and the strides used can affect the output of a convolutional layer. However, it is not easy to determine the relationship between these variables [38]. We use Equation (Equation 1) to illustrate the relationship between these variables as suggested in [38],
(1)$$out=\lfloor \frac{y+2p-k-(k-1)(d-1)}{s}\rfloor +1$$

Here, *y* is the input shape, *p* is padding, *k* is the kernel size, *d* is the dilation that increases the respective field of output without changing the kernel size and *s* indicates how much the kernel moves vertically and horizontally through the input matrix.

Changing the patch size in Convolutional Neural Network (CNN) can offer robust performance regardless of dataset variation [39]. We propose the NAP augmentation module, with ideas inspired by the CNN architecture (Equations (Equation 2)–(Equation 4)). Our padding (p) value is 0 since we generate patches from a given image by applying pixel-wise convolution. The dilation (d) value can be utilized to simplify the CNN-based architecture. On the other hand, ref. [40] proposed a method for training CNN-based architecture that uses dilation values, without considering coverage or resolution loss. However, since our NAP augmentation module performs pixel-wise convolution to extract patches from a given image before training the CNN-based architecture, the patches could be distorted. Therefore, our dilation (d) value is set to 0. When we substitute all these numbers into Equation (Equation 1), we can see that our Equations (Equation 2) and (Equation 3) are simplified versions of Equation (Equation 1). We introduce three new equations derived from Equation (Equation 1) [38] to accomplish the NAP algorithm. The NAP augmentation module is shown at the bottom right corner of Figure 2. A full procedure is given in the Algorithm 1.
(2)$$p_{h}=\left\lfloor \frac{y-k}{s}\right\rfloor +1$$
(3)$$p_{l}=\left\lfloor \frac{y-k}{s}\right\rfloor +1$$
(4)$$N=p_{h}\times p_{l}$$

**Algorithm 1:**NAP Algorithm

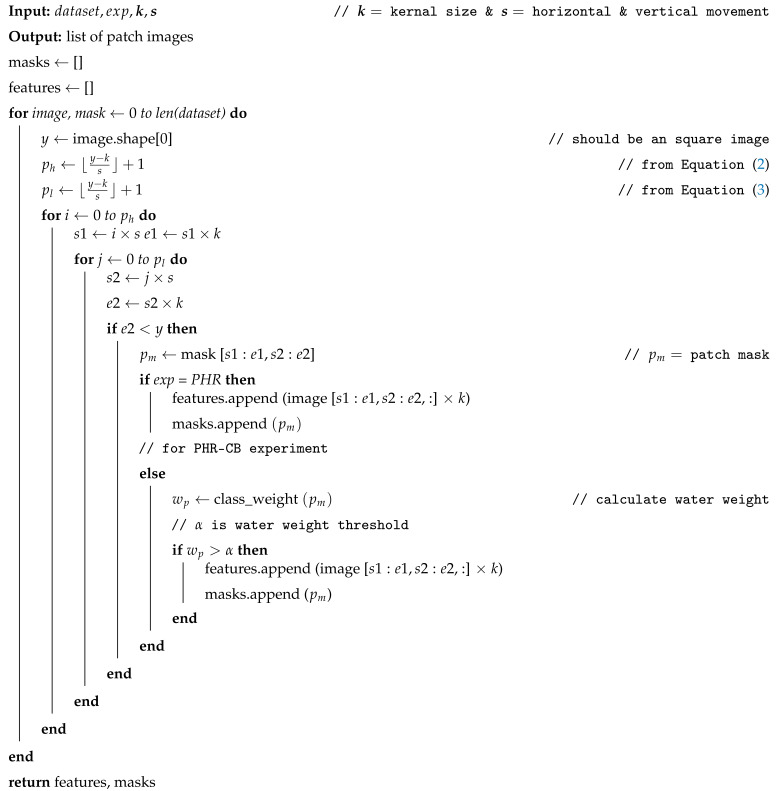



Here, $p_{h}$ represents the number of times the unit kernel moves horizontally and $p_{l}$ represents the number of times the unit kernel moves vertically, dictated by the image dimension, patch size and stride. *N* represents the number of patches generated from a given image.

### 3.3. Proposed FAPNET Model

The architecture of our proposed FAPNET model is illustrated in Figure 3. The three input channels consist of the VV, VH and DEM. The DFM processes these input channels and identical-sized feature maps are generated and concatenated before passing as input into our encoder–decoder network for pixel-level labeling. In our encoder–decoder architecture, we do not utilize any pre-trained model weight. However, our NAP augmentation module, as stated in Section 3.2, enhances our model’s performance. The entire network configuration is described in Table 2 and Table 3, using various experimental setups, which will be discussed in Section 4. Tables include the kernel width, kernel height and number of kernels for each convolutional layer, as well as the output sizes of the feature maps. A summary of our model algorithm is given in Algorithm 2.
**Algorithm 2:**Proposed FAPNET Model Algorithm
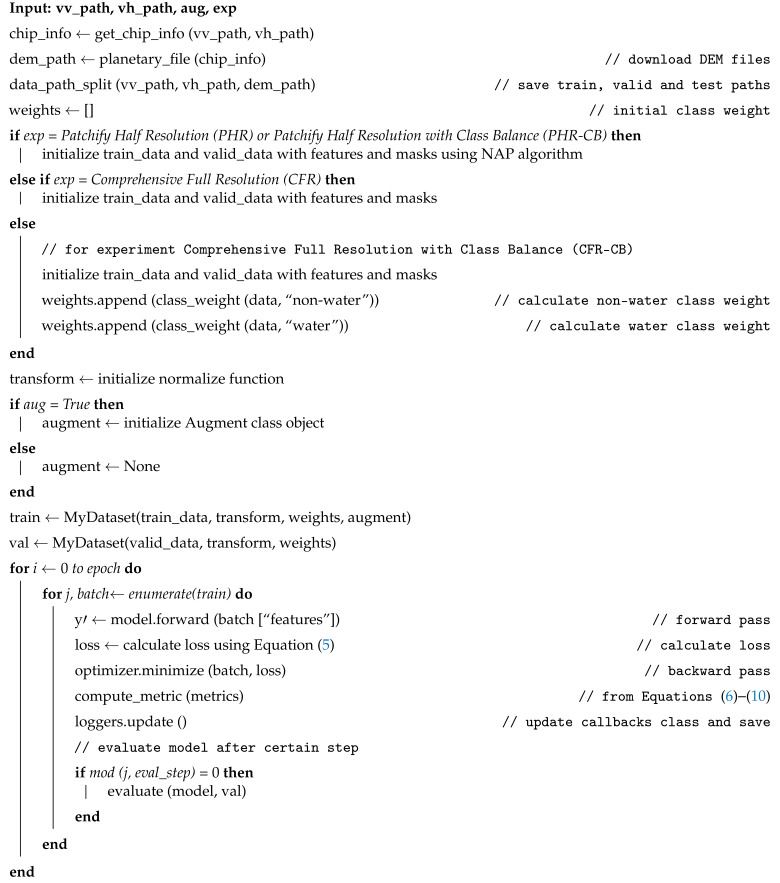


The model uses two blocks: a downblock and an upblock to down-sample and up-sample. The first layer, downblock-1 and upblock-1, is made up of 2-kernel (3×3) convolutional networks with the same padding size, followed by a ReLU activation function and a dropout layer. Downblock-2 through downblock-6 have a similar structure to downblock-1, but are followed by a 2×2 max-pooling layer to downsize the feature height and width by half. Upblock-1 is constructed similar to downblock-1, except upblock-1 is followed by a 2×2 convolutional layer with same padding size and SoftMax activation. In addition, each layer’s downblock has a skip connection that connects it to its equivalent upblock.

Each convolutional layer of a single downblock contains the same number of kernels and is multiplied by two for the following downblocks up to downblock-4. In downblock-5 and downblock-6, the second convolutional layer kernel numbers are decreased by one-fourth and half of downblock-4, respectively. On the other hand, the number of kernels in upblock-6 to upblock-4 is similar, while the number of kernels in upblock-3 to upblock-1 is divided by 2 with regard to the preceding upblock, as shown in Table 2. Our proposed FAPNET middleblock is made up of a max-pooling layer having two 3×3 convolutional networks with ReLU activation function and a dropout layer, all of which are connected by 2×2 convolutional transpose with the same padding size.

### 3.4. Hyperparameter Tuning

Hyperparameters have the potential to directly influence the training algorithm’s behavior. Choosing optimal hyperparameters has a significant impact on the performance of the trained model. Traditionally, the optimal hyperparameter values for a given dataset were determined manually. Researchers usually lean on their prior experience in training machine learning algorithms to choose these parameters. The disadvantage is that the optimal setting for hyperparameters used to solve one problem may not be optimal for a different problem, as these values vary between datasets and even image categories. Consequently, it is challenging to establish the hyperparameter values only based on past experience [41].

To improve our model dropout and number of kernels in each layer, we utilize the auto Keras framework [42]. The framework uses Bayesian optimization to guide through the search space by identifying the most promising operations each time, resulting in an efficient neural architecture search for hyperparameter tuning. It also utilizes an edit-distance neural network kernel to determine the number of operations needed to transfer from one neural network to another [42]. After tuning, we obtain the optimal number of kernels in each layer, as shown in Table 2 and Table 3, where in each conv2d layer ($Conv2D_{1}$ and $Conv2D_{2}$ column), inside the parenthesis it indicates the feature’s height, width and the number of the kernels, respectively.

When gradient descent hits a cost surface minimum, parameter values might fluctuate around the minimum. To counteract this, gradient calculation can be optimized by reducing the learning rate [43]. To resolve the issue, we apply a learning rate scheduler that reduces the learning rate after certain epochs.

### 3.5. Loss Function

Focal Loss (FL) [44] is a BinaryCross-Entropy variation. It reduces the importance of easy examples, allowing the model to concentrate more on learning difficult instances. It performs well in circumstances where the classes are very imbalanced. By using Equation (Equation 5), we can calculate FL.
(5)$$FL\left(y,y^{\prime }\right)=-\alpha {\left(1-y^{\prime }\right)}^{\gamma }y\log\left(y^{\prime }\right)-\left(1-\alpha \right)y^{\prime \gamma }\left(1-y\right)\log\left(1-y^{\prime }\right)$$
where *y* is the ground truth and $y^{\prime }$ is the model prediction. The α value is 0.25 and can be treated as a hyperparameter. γ can also be treated as a parameter, but as per [44], setting its value to 2 provides the best result.

### 3.6. Evaluation Metrics

Since we perform a binary segmentation, there are two possible outcomes: positive or negative, which correspond to water or background area, respectively. True Positive (TP) is a segmentation result that shows water is identified correctly. True Negative (TN) indicates that background pixels are classified correctly. False Positive (FP) refers to background incorrectly identified as water. Water pixels are classified incorrectly as background if False Negative (FN) occurs. Several measures can be used to assess segmentation performance. In general, we want to know the percentage of pixels that have been correctly identified, regardless of their class. Precision, recall, F1 [45], the Mean Intersection Over Union (MeanIoU) [46] of the water class and Dice Coefficient [47] scores are often used in the literature.
(6)$$Precision=\frac{TP}{TP+FP}$$
(7)$$Recall=\frac{TP}{TP+FN}$$
(8)$$F1=2\times \frac{Precision\times Recall}{Precision+Recall}$$

For the segmentation evaluation, the MeanIoU [46] score is a commonly used performance metric. The MeanIoU [46] measure is given in Equation (Equation 9).
(9)$$IoU=\frac{TP}{TP+FP+FN}$$

The Dice Coefficient [47] is a broadly used indicator in the computer vision domain to determine the similarity of two images [48]. It has evolved into a loss function called Dice Loss [47].
(10)$$DC\left(y,y^{\prime }\right)=\frac{2\times |y\cap y^{\prime }|+1}{y+y^{\prime }+1}$$
where *y* is the ground truth and $y^{\prime }$ is the model prediction. In the numerator and denominator, 1 is added so that it does not become undefined if $y=y^{\prime }=0$

## 4. Performance Evaluation

We used three different evaluating metrics—quantitative, qualitative and memory-time efficiency analysis—to verify the effectiveness of our NAP augmentation technique and DFM, integrated in the proposed FAPNET architecture. We will describe our four experimental designs in this section, along with the results that they produced. Even though we output the training, test and validation results, we focus on the test dataset for performance evaluation because the test data is completely unseen during training. FAPNET is compared with other segmentation models UNET, VNET, DNCNN, UNET++, U2NET, ATTUNET, FPN and LINKNET. Based on the evaluating metrics, quantitative, qualitative and memory-time efficiency analyses, FAPNET consistently outperforms other models. Since the parameter settings and preprocessing steps are the same across all experimental setups (CFR, CFR-CB, PHR, PHR-CB), the comparison is fair. The quantitative analysis described in Section 4.1 includes an explanation of the experimental setups. In addition to comparing our outcomes with those of the competitors (Section 4.9), we also analyze the significance of the DFM in Section 4.8.

### 4.1. Quantitative Analysis

There are a number of evaluation metrics as described in Section 3.6 above. Since MeanIoU combines precision and recall statistics into a compact score [46], it has become increasingly used in the literatures over the past decade. Therefore, in our experiments, we will focus on using MeanIoU to analyze the test dataset results, which are generated from four setups: CFR, CFR-CB, PHR and PHR-CB. Figure 4 shows an overview of the four experimental workflows. We focus on the MeanIoU score under the testing dataset column because it reflects the outcome after model training.

### 4.2. CFR

In this setup, the data is passed through the DFM, similar to the state-of-the-art methods mentioned in the “Map flood-water from Radar Imagery” competition. All images/chips have the same resolution (512×512). We use data augmentation and data normalization. Pixel information is given in Table 4, which will be required in Section 4.3, CFR-CB experiment and the results of the CFR experiments are shown in Table 5. FPN is created to comprehend LiDAR point cloud [49] and it can be seen that the FPN [49] model is the second best. Our FAPNET performs best overall and has the best MeanIoU of 0.8544 in the testing dataset.

### 4.3. CFR-CB

In the CFR setup, results may be affected by class imablance, which can lead to unanticipated inaccuracy or significant adverse effects in data analysis, particularly during the classification tasks [50]. From Table 4, we can see that background (no water) populates 86% in the testing dataset and in our CFR setup above, we did not address the data imbalance problem. In this CFR-CB experiment, we introduce a re-weighting technique to adjust the imbalance ratio by inversely multiplying our mask with the class weights shown in Table 4. Loss balancing is also obtained as a result of this re-weighting method [51].

The results of the CFR-CB experiment are shown in Table 6. After balancing the dataset by weight, most models’ scores rise by at least 1% to 6% in general compared to Table 5. However, for the DNCNN [52] and ATTUNET [53] models, the opposite occurs. their model architecture may be sensitive to the class balance weight. FPN remains the second best. However, LINKNET has the biggest improvement, achieving a MeanIoU of 0.8638 from 0.8021, overtaking UNET as the third best. On the other hand, FAPNET surpasses other models in terms of the MeanIoU on the testing dataset with a score of 0.8544.

### 4.4. PHR

In our previous two experiments, we use the classical data augmentation technique and class balance in the CFR-CB setup. However, in this PHR experiment, we use the NAP augmentation module described in Section 4.5 without class balancing. We generate image patches with patch size (k)−256 and stride (s) value 64. As mentioned before, the image shape in the dataset is 512×512 and by passing a single image through the NAP augmentation module, we get 25 images of dimension 256×256.

Table 7 shows that nearly all models (except U2NET) give inferior scores compared to Table 5 and Table 6. Note that in this setup, we generate patches from the dataset, producing around 10K patches, which also creates many patches that do not have any water pixels, leading to large variations between classes. As a result, the models are biased towards the non-water class, affecting the outcome accuracy. Our FAPNET appears more sensitive to this class imbalance compared to other models. In order not to introduce bias in the evaluation, we design PHR-CB, taking class balance into consideration.

### 4.5. PHR-CB

We can see from the PHR experimental setup that without class balancing, the outcome can be biased in favor of models that are “data hungry” or have a high number of parameters. In the PHR experiment, we generate approximately 10K patches from 433 training images, but it leads to a data imbalance problem because the NAP augmentation approach generates many patches that do not contain any water or have a low proportion of water, due to the image content of the “Map flood-water from Radar Imagery” competition dataset. We solve this issue by rewriting Equation (Equation 4) as shown in Equation (Equation 11) for this PHR-CB experiment. In Equation (Equation 11), we set α to 0.20, adjusting the water and non-water proportion back to the proper class distribution.

The number of alleviated patches after using the NAP augmentation approach is denoted by α in Equation (Equation 11).
(11)$$N=p_{h}\times p_{l}-\alpha$$

Table 8 shows that overall our proposed FAPNET model outperforms other MeanIoU scores in the test dataset. It even shows better results than its performance in other experiments, including CFR-CB, which is also class balance. This is attributed to changing the NAP augmentation module Equations (Equation 4)–(Equation 11), which reduces the high class variation. It is worth pointing out that compared to all other models and all experimental setups, FAPNET provides the best MeanIoU scores of 0.9514, 0.8960 and 0.8706 on training, validating and testing samples, respectively, in the PHR-CB experiment.

Based on the quantitative analysis, we can conclude that our model is sensitive towards class imbalance and the PHR-CB experimental setup, which incorporates the NAP augmentation and class-balance technique, is most suitable and unbiased in terms of addressing the class imbalance for all models. In Figure 5, we compare the training, validation and testing dataset MeanIoU score for the experiments across the models.

### 4.6. Qualitative Analysis

We present in this section our qualitative analysis, which also demonstrates that our approach outperforms other models. From our testing dataset, we randomly selected three predictions, which are shown in Figure 6, where the first four rows represent VV, VH, DEM and Ground Truth (GR), respectively. The remaining rows show model predictions. Three examples are given and the results from the four experimental setups are shown under each example. In the CFR experiment, it is noticeable that U2NET [57], DNCNN [52], ATTUNET [53], UNET++ [56] and VNET [55] cannot correctly predict water pixels in all three examples. In contrast, our FAPNET, UNET [54], FPN [49] and LINKNET [58] can accurately predict more water pixels than other models. In the second example, we can observe that our model predicts water pixels more accurately than other models. UNET [54], FPN [49] and LINKNET [58] are unable to predict the top left corner of the expected mask and are less accurate than our model (as shown in the final row, fifth column). Another example of this behaviour can be seen in the last row, ninth column (top center and right side of the predicted mask). We can also analyze each experimental setup in Figure 6. Overall, our proposed FAPNET outperforms all other models in the CFR experimental setup. The majority of the models perform well in the CFR-CB experiment, capturing more water distributions.

In the PHR experiment, we observe that FPN [49], UNET [54], LINKNET [58] and our proposed FAPNET models (all performed well in the CFR and CFR-CB experiments) tend to diverge from the ground truth. The dataset size in this experiment was approximately 10k and there were many patches without water pixels, as described in Section 4.5, which is attributed to the decreased model performance due to class imbalance. In contrast, the prediction of U2NET [57] improves, because it can extract more features from larger datasets. In addition, a closer look at the third example shows that, with the exception of FPN [49], all other model predictions are lower in this experiment. We can conclude from this finding that highly imbalanced data affect accurate model prediction. Increasing the number of training samples can also introduce high class variance. That is the reason we introduce the PHR-CB experimental setup.

PHR-CB is the most successful experimental design since every model prediction increased significantly in terms of qualitative analysis. As can be seen in Figure 6, the predictions of U2NET [57], DNCNN [52], ATTUNET [53], UNET++ [56] and VNET [55] improve substantially in each example. However, note that some models have a tendency to predict noise and our FAPNET model predicts a certain amount of noise in the first and third examples, but the noise is considerably less than other existing models. In Figure 6, the first example, we can observe that the FAPNET model accurately predicts water distribution that other models overlook. In Figure 6, the second example, our model’s prediction is the most accurate among all models, and in the third example, FAPNET predicts almost all of the water pixels with a very low level of noise. Some small waterbody pixels are not properly identified in any of the models’ predictions, whereas FAPNET detects more small water areas as well as larger area, especially in the PHR-CB experiment.

Overall, the results demonstrate that the PHR-CB and CFR-CB experiments, with class balancing, are the best setup for qualitative performance evaluation. The PHR-CB experiment is the best for quantitative evaluation as all models increase their performances in general, with our FAPNET model outperforming others.

### 4.7. Memory-Time Efficiency Analysis

In this Section, we compare the model’s GPU, RAM and memory usage, as well as the run-time for different batch sizes and epochs. There are trade-offs between accuracy, memory usage and time performance when evaluating a model. A model’s number of parameters may also indicate whether it is lightweight. When developing the Refactor Equation (Equation 12), we chose to deploy U2NET [57] parameters because of its large parameter values as the numerator. If the refactor value is lower than the others, we know that the model is lighter than the others.
(12)$$refactor=\frac{parameters}{highest\_parameters}$$

From the refactor value, we can see that our proposed model FAPNET contains more than three times as many parameters as UNET [54], as shown in Table 9. Yet its performance in terms of memory usage and training time is comparable to UNET [54]. Models with huge amount of parameters and complicated architectures tend to extract more features, e.g., U2NET [57] and ATTUNET [53], which increases training time and resource usage, as shown in Table 9. In addition, models with complicated architectures are data-hungry, as observed in the PHR experiment, where the U2NET [57], VNET [55] and UNET++ [56] models perform well as the dataset is six times larger. On the other hand, models with simple architectures, such as UNET [54] and our proposed model FAPNET, can perform well in all experiments in terms of time performance, memory usage and accuracy. Overall, among the models with a simple design, our model outperforms UNET [54], as demonstrated by the quantitative and qualitative evaluations.

### 4.8. Significance of DFM Module

From the above experimental analysis, it can be seen that our FAPNET model yields the best results in the PHR-CB experimental setup when using the DFM module. However, to assess the importance of VV, VH and DEM features independently, we need to conduct a new experiment using each feature as input, followed by a result comparison using qualitative, quantitative and memory-time efficiency analyses.

In this section, we discuss the importance of the DFM module by conducting more experiments, in which we employed different combinations of model input (VV, VH, DEM and DFM) and compared the prediction accuracy. We also checked the significance of each feature of the DFM module and discovered which feature has more influence over flood-water prediction. We used the FAPNET model and the PHR-CB experimental setup to compare the effectiveness of individual input features based on qualitative, quantitative and memory-time efficiency analyses. Table 10 shows the quantitative analysis for each input feature, where DFM outperforms others with a MeanIoU score of 0.8706, while the VV, VH and DEM features obtain MeanIoU scores of 0.5405, 0.6063 and 0.5404, respectively.

The results show that VH has more influence than VV for flood-water prediction, while DEM only provides high-level features. Figure 7 shows the predictions from the VV and VH features are too noisy, while the DEM prediction is almost null. However, the DFM prediction is less noisy and almost similar to the ground truth. This is because we combine scaling with normalization in the DFM module, which enhances the model performance. We also performed memory-time efficiency analysis. Table 11 shows that DFM takes longer time to train the model than other feature models. While the CPU consumption is similar, the GPU consumption in DFM is higher than other feature models. Overall, FAPNET provides the best performance in quantitative and qualitative analysis, but has a longer training time and memory consumption.

### 4.9. Comparison with Competition Results

The best-performing model in the Microsoft AI on Earth Competition achieved a MeanIoU score of 0.8094, showing a significant increase over the benchmark result (0.4357). The second and third positions are slightly less than the top position—0.8072 and 0.8036, respectively. All suggested architectures are able to accurately recognize the larger water surface area. However, they all use segmentation techniques without taking into account the relevance of tiny water pixels (small salient features). Moreover, no data preparation or data augmentation methods were used on top of the UNET-based benchmark solution. As polarized SAR data make available signals from a wide variety of landscapes and plant types, all winning approaches from the competition utilized the Planetary Computer STAC API to incorporate supplementary data from the NASADEM and/or the European Commission’s Joint Research Centre (JRC) global surface water data in their models, leading to a better understanding of the geographic area’s natural topography. We designed a new FAPNET architecture for better detection of tiny water pixels, which are often missed by existing models. Our DFM is used for noise suppression and NAP augmentation is used to boost both quantitative and qualitative performance. Figure 8 shows that FAPNET performs better than the top scores in the competition.

## 5. Discussion

Our motivation to introduce the FAPNET architecture came from the UNET [54] and our model contains three times as many parameters as UNET [54]. Though the FAPNET refactor 0.13 is almost the same as other existing models, it takes less time for training, as reported in Table 9. Hyperparameter tweaking also makes our model more versatile in terms of extracting more salient features. Table 9 demonstrates that, although having more parameters than UNET [54], our FAPNET model uses the same amount of time to train and outperforms other models with the same number of parameters, i.e., FPN [49], UNET++ [56] and LINKNET [58]. In general, the class balance in the CFR-CB experiment enhances model performance for all models, compared to the CFR experiment. This indicates that data preprocessing and class balance are essential factors for increasing model performance. However, we also found that data augmentation incorporated in our proposed NAP augmentation module has both pros and cons that differ between models and experimental setups. In the PHR experiment, our FAPNET model is underrated in comparison to other models. Yet, the class balance setup in the PHR-CB experiment gives it an advantage over other models and experiments, leading to the best MeanIoU scores in training, validating and testing datasets. Overall, combining the PHR-CB experimental setup with our proposed FAPNET model outperforms all other existing models in terms of quantitative, qualitative and memory-time efficiency performance.

For two reasons, training data collection has increasingly become a major concern. First, as machine learning is becoming more popular and more commonly used in industry development and academic research, new applications are emerging that require greater varieties of labeled data. Second, unlike traditional machine learning algorithms, deep learning algorithms explore complex feature representations, cutting costs on feature engineering, which often requires more labeled data [59]. The dataset we used in our experiments has some flaws, as stated in Section 2.1. Fixing these issues will likely affect the model performance but might produce better results. Model performance may be affected when shifting from traditional data augmentation strategies to cutting-edge techniques like feature transplanting, style transfer and GAN-generated augmented data. As discovered in our PHR-CB experiment, input shape and class balance can affect model performance. Hence, it is worthwhile to investigate a broader scope of possibilities, e.g., testing whether model performance is affected by varying the input shape. Class balance has a direct impact on loss calculation and it is an important aspect in deep learning model training. Note that adopting a custom loss function may influence the model’s performance. Additionally, we should be able to enrich our model by employing the hyperparameter tuning approach with regards to dropout and the number of kernels. Performance will likely be improved if we adjust other hyperparameters using various optimization methods.

UNET was initially invented to address medical image segmentation problems. Later, researchers applied the UNET architecture on pixel-level classification in numerous applications. Radar signals have complex hyperspectral characteristics beyond the visible spectrum, which often creates challenges in the analysis process. To the best of our knowledge, hyperspectral image processing using deep learning is still inadequately explored in the research community. We preprocessed hyperspectral images and introduced our FAPNET model, with more fine-tuned parameters than other existing models. The results outperformed related works, especially when combining with the NAP augmentation algorithm and DFM module. Experimental results show that FAPNET performs better than UNET with comparable training time. We believe that FAPNET can perform well on optical images in terms of quantitative, qualitative and memory-time efficiency analysis, because the RGB bands (long, medium and short wavelengths) are a part of the electromagnetic spectrum. Since the focus of this work is for flood mapping-related analysis based on the invisible band features of the spectrum, we will leave the optical images to future work when studying other applications.

Another idea for future work is to analyze multiple features simultaneously inside our DFM and increase the number of channels (e.g., 4–6 channels instead of 3 channels) to accommodate both the visible RGB and invisible radar signal bands. We will be able to compare the results of this neural net architecture with the outcome from transfer learning discussed above. For now, FAPNET outperforms other models for detecting waterbodies using the invisible bands as input.

## 6. Conclusions

In this paper, inspired by the widely adopted deep learning techniques in satellite remote sensing, we propose a variety of novel methods for extracting waterbody from satellite imagery. In comparison to prior models, our proposed FAPNET model utilizes fewer parameters while achieving superior qualitative and quantitative results. By introducing the DFM module and NAP augmentation algorithm in the data preprocessing stage, our FAPNET model delivers better performance and reduces computational overhead compared with related works. Our DFM combines and normalizes feature bands, while the NAP augmentation algorithm creates image patches in the required resolution without altering the pixel values. We compared our NAP algorithm with commonly used augmentation techniques (rotation, flip, etc.) and found that by adopting NAP, the model could identify even a tiny water surface. Our proposed FAPNET, which incorporates DFM and the NAP augmentation method, outperforms the state-of-the-art methods in the PHR-CB experimental configuration based on the MeanIoU score, memory consumption and training execution time. The tuning of the hyperparameters helps in choosing optimal parameter values. Qualitative evaluation shows that our model outperforms other data preprocessing methods and deep learning frameworks like UNET when it comes to detecting the presence of water. We anticipate that our model will be able to identify even the tiniest waterbody; however, this is balanced by the fact that it also forecasts a fair amount of background noise. The traditional UNET-based model structure works well in detecting big water regions. Nevertheless, we observed that an augmentation mechanism may assist deep learning models in recognizing tiny areas with less noise (encoder–decoder-based model). Transforming from older data augmentation methods to more modern techniques could further enhance model efficiency. We hope to explore more advanced techniques in future work, including feature transplanting, style transfer and generative adversarial network (GAN). In our PHR-CB experiments, we discovered that the model’s performance was affected by the diverse input feature shapes and class re-weighting. We will look into a new model layer design that can accommodate a wide variety of input shapes. For class re-weighting, we will look into new loss functions.

## Figures and Tables

**Figure 1 sensors-22-08245-f001:**
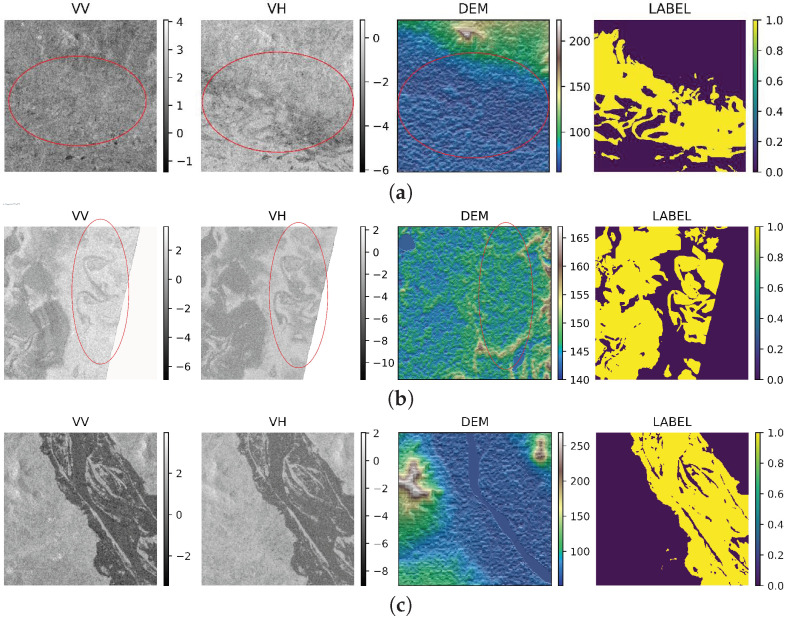
VV and VH are plotted in grayscale mode, while DEM and Ground Truth (GR) are plotted using Earthply package and matplotlib, respectively. (**c**) An ideal case with consistent water features and mask (label); (**a**) shows that features and mask have significant variation; and (**b**) illustrates that some parts of the mask image do not match features well, especially when comparing with DEM. In the fourth column, yellow indicates water label and purple denotes background. Unassigned pixels are labeled as non-water in the mask, e.g., (**c**) bottom right corner.

**Figure 2 sensors-22-08245-f002:**
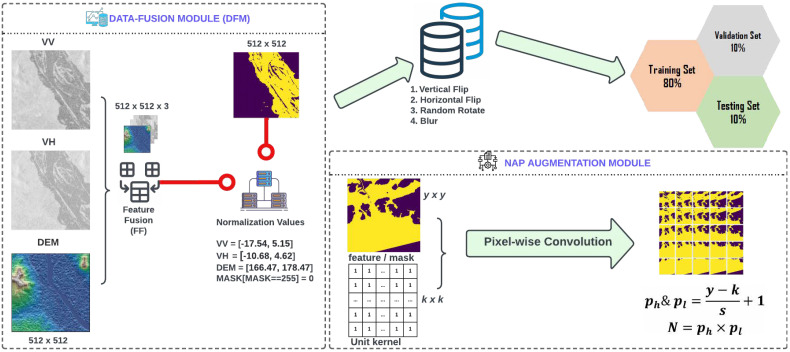
The data is passed to DFM for normalization. The normalized data is processed by the data augmentation library and divided into training, validation and testing samples. In the PHR and PHR-CB experiments, the data is passed through the NAP augmentation module after DFM but for the CFR, CFR-CB experiments, NAP is not required.

**Figure 3 sensors-22-08245-f003:**
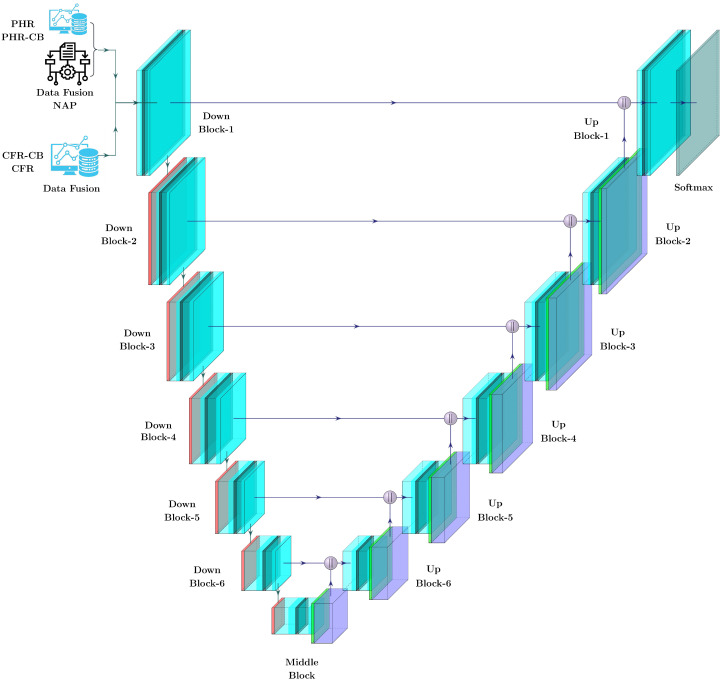
The FAPNET model takes the DFM output and passes data through or skips the NAP augmentation module, depending on the experimental setup. Each layer’s downblock output and the previous layer’s upblock are concatenated and the final pixel-wise prediction is provided by CNN with softmax. Between the CNN layers, each down-block and up-block has a dropout layer with a value of 0.2.

**Figure 4 sensors-22-08245-f004:**
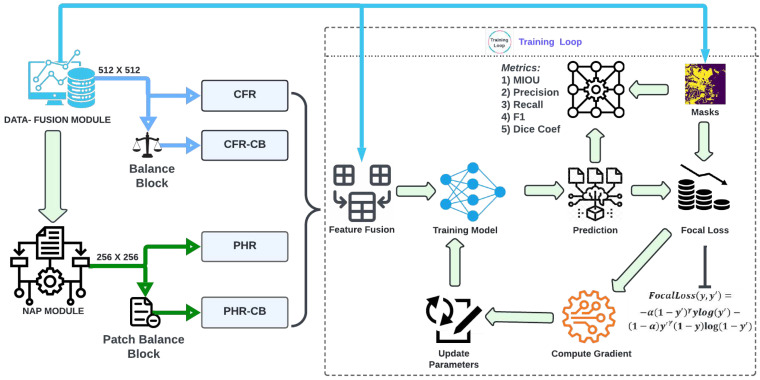
Pass Feature Fusion (FF) and Mask to the Training Loop based on the experimental setup, using either DFM or a combination of DFM and NAP augmentation module. The loss is calculated using Focal Loss and the model parameters are updated using Gradient. The model is evaluated using MeanIoU as it is most commonly used by researchers. However, for reference purposes, we also report the Precision, Recall, F1 and Dice Coefficient.

**Figure 5 sensors-22-08245-f005:**
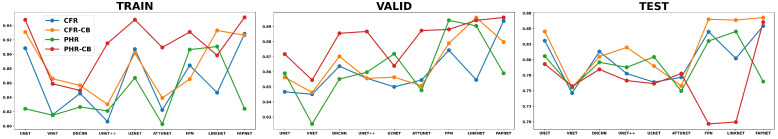
Experimental result comparison on the datasets using the MeanIoU score. In the CFR-CB and PHR-CB experiments (both with class balance), our FAPNET gives better MeanIoU scores than the other models and other experiments with class imbalance samples.

**Figure 6 sensors-22-08245-f006:**
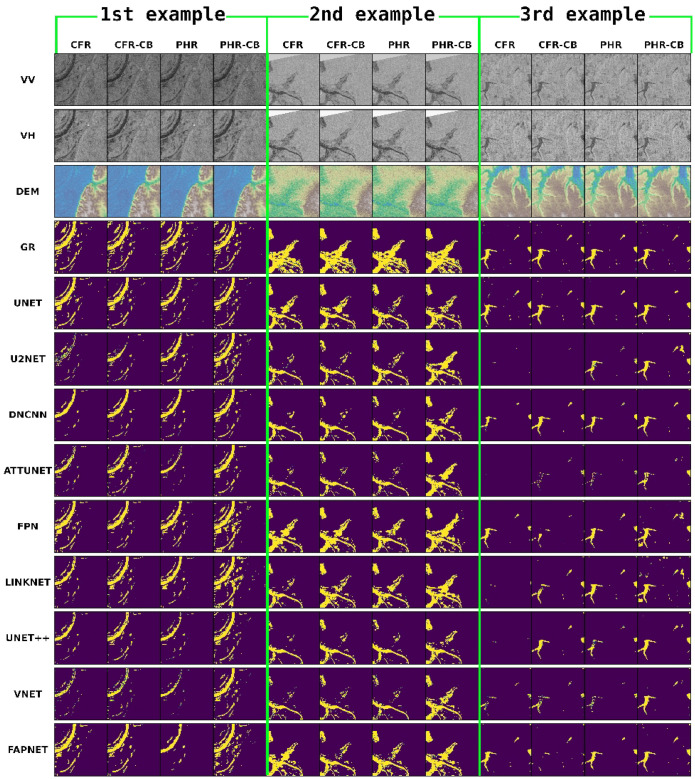
First four rows show VV, VH, DEM and GR, respectively. VV, VH are plotted in grayscale mode, while DEM and GR are plotted using the Earthply package and matplotlib, respectively. A DEM represents ground elevations, where a color indicates the elevation at the pixel location. The color scale ranges from the minimum elevation value (blue) to the maximum elevation value (brown). Green and Yellow represent the elevations in between. We focus on blue for our water-body detection task. The remaining rows indicate different model predictions and the last row is our proposed FAPNET model prediction from the test dataset. Three randomly selected examples are presented for comparison between existing models and our proposed FAPNET model. Each column under an example represents an experimental setup described in Section 4.2, Section 4.3, Section 4.4 and Section 4.5, respectively.

**Figure 7 sensors-22-08245-f007:**
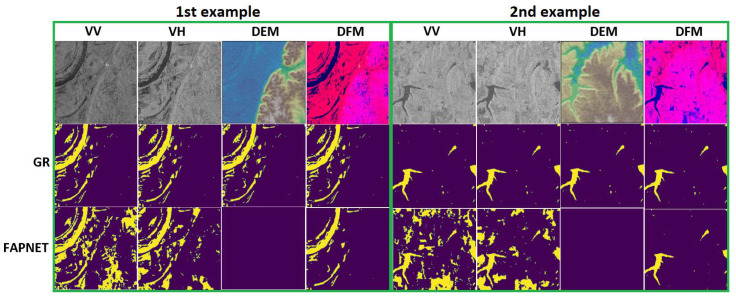
First, second and third row indicate the input feature, ground truth and predicted mask by FAPNET, respectively, for the PHR-CB experiment. VV and VH are plotted in grayscale mode, while DEM, GR, DFM and FAPNET predictions are plotted using Earthply package and matplotlib, respectively. A DEM represents ground elevations, where a color indicates the elevation at the pixel location. The color scale ranges from the minimum elevation value (blue) to the maximum elevation value (brown). Green and Yellow represent the elevations in between. We focus on blue for our water-body detection task. Pink/red and dark blue color in the DFM indicate background and water respectively Two examples are selected which we compared earlier, in Figure 6.

**Figure 8 sensors-22-08245-f008:**
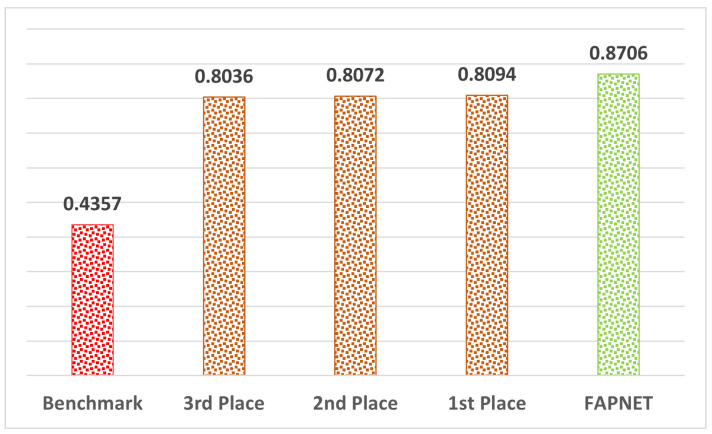
Performance comparison with the competition’s 1st, 2nd and 3rd outcomes.

**Table 1 sensors-22-08245-t001:** Over five years, 542 chips or 1084 images from 13 geographic regions were acquired. Each chip contains VV and VH information.

Location	No of Chip	Month	Year
Bolivia	15	February	2019
Cambodia	30	August	2018
Ghana	53	September	2018
India	68	August	2016
Nigeria	18	September	2018
Pakistan	28	June	2017
Paraguay	66	October	2018
Slovakia	65	October	2020
Somalia	26	May	2018
Spain	30	September	2019
Sri Lanka	42	May	2017
UK	32	February	2019
USA	69	May	2019
Total	542		

**Table 2 sensors-22-08245-t002:** FAPNET model layer construction for the **CFR** and **CFR-CB** experiments.

Layers	Conv2D${}_{1}$	Conv2D${}_{2}$	Padding	Non-Linearities
DataFusion	(512, 512, 3)	-	-	-
downblock-1	(512, 512, 16)	(512, 512, 16)	Same	ReLU
downblock-2	(256, 256, 32)	(256, 256, 32)	Same	ReLU
downblock-3	(128, 128, 64)	(128, 128, 64)	Same	ReLU
downblock-4	(64, 64, 128)	(64, 64, 128)	Same	ReLU
downblock-5	(32, 32, 256)	(32, 32, 32)	Same	ReLU
downblock-6	(16, 16, 512)	(16, 16, 64)	Same	ReLU
middleblock	(8, 8, 1012)	(8, 8, 256)	Same	ReLU
upblock-6	(16, 16, 128)	(16, 16, 128)	Same	ReLU
upblock-5	(32, 32, 128)	(32, 32, 128)	Same	ReLU
upblock-4	(64, 64, 128)	(64, 64, 128)	Same	ReLU
upblock-3	(128, 128, 64)	(128, 128, 64)	Same	ReLU
upblock-2	(256, 256, 32)	(256, 256, 32)	Same	ReLU
upblock-1	(512, 512, 16)	(512, 512, 16)	Same	ReLU
Conv2D	(512, 512, 2)	-	-	Softmax

**Table 3 sensors-22-08245-t003:** FAPNET model layer construction for the **PHR** and **PHR-CB** experiments.

*Layers*	Conv2D${}_{1}$	Conv2D${}_{2}$	Padding	Non-Linearities
DataFusion+Nap	(256, 256, 3)	-	-	-
downblock-1	(256, 256, 16)	(256, 256, 16)	Same	ReLU
downblock-2	(128, 128, 32)	(128, 128, 32)	Same	ReLU
downblock-3	(64, 64, 64)	(64, 64, 64)	Same	ReLU
downblock-4	(32, 32, 128)	(32, 32, 128)	Same	ReLU
downblock-5	(16, 16, 256)	(16, 16, 32)	Same	ReLU
downblock-6	(8, 8, 512)	(8, 8, 64)	Same	ReLU
middleblock	(4, 4, 1012)	(4, 4, 256)	Same	ReLU
upblock-6	(8, 8, 128)	(8, 8, 128)	Same	ReLU
upblock-5	(16, 16, 128)	(16, 16, 128)	Same	ReLU
upblock-4	(32, 32, 128)	(32, 32, 128)	Same	ReLU
upblock-3	(64, 64, 64)	(64, 64, 64)	Same	ReLU
upblock-2	(128, 128, 32)	(128, 128, 32)	Same	ReLU
upblock-1	(256, 256, 16)	(256, 256, 16)	Same	ReLU
Conv2D	(256, 256, 2)	-	-	Softmax

**Table 4 sensors-22-08245-t004:** Pixel-wise distribution across the training, validation and testing dataset, with class weights listed in the last column.

Class	Training	Validation	Test	Class Weights
**Water**	15,804,563	2,611,509	1,941,970	0.14
**Background**	97,703,789	11,544,267	12,475,950	0.86
**Total**	113,508,352	14,155,776	14,417,920	1.0

**Table 5 sensors-22-08245-t005:** **CFR** experimental results on floodwater dataset [8]. Average result was taken after 150 epochs with a ±2.50% error rate. **Green**, **Yellow** and **Blue** in the MeanIoU Test dataset column indicate first, second and third best, respectively.

	Focal Loss			MIoU			Dice Coff			F1			Precision			Recall		
Model	Trn	Val	Tst	Trn	Val	Tst	Trn	Val	Tst	Trn	Val	Tst	Trn	Val	Tst	Trn	Val	Tst
**UNET** [54]	**0.0726**	0.0788	**0.0767**	**0.9084**	0.8467	**0.8306**	**0.9823**	0.9497	**0.9567**	**0.9527**	**0.8792**	**0.8455**	**0.9635**	0.8845	0.8961	**0.9478**	**0.8761**	**0.8370**
**VNET** [55]	0.0779	0.0782	0.0803	0.8158	0.8451	0.7463	0.9481	0.9468	0.9353	0.8508	0.8478	0.7402	0.8827	**0.9385**	**0.8995**	0.8460	0.8147	0.7119
**DNCNN** [52]	0.0761	0.0769	0.0775	0.8455	**0.8637**	0.8132	0.9582	**0.9536**	0.9495	0.8414	0.8005	0.7915	0.8798	0.8424	0.8932	0.8377	0.8057	0.7723
**UNET++** [56]	0.0783	**0.0763**	0.0791	0.8063	0.8557	0.7777	0.9478	0.9505	0.9399	0.7754	0.7510	0.7444	0.8247	0.8065	0.8577	0.7941	0.7470	0.7365
**U2NET** [57]	**0.0728**	0.0779	0.0796	**0.9071**	0.8500	0.7639	**0.9754**	0.9521	0.9389	0.8437	0.7868	0.7463	0.8888	0.8623	0.8866	0.8420	0.7779	0.7320
**ATTUNET** [53]	0.0770	0.0777	0.0796	0.8225	0.8545	0.7717	0.9546	0.9511	0.9397	0.7835	0.7565	0.7393	0.8469	0.8048	0.8767	0.7933	0.7576	0.7224
**FPN** [49]	0.0738	**0.0748**	**0.0769**	0.8845	**0.8743**	**0.8450**	0.9699	**0.9558**	**0.9578**	**0.8880**	**0.8538**	**0.8541**	**0.9149**	**0.8873**	**0.9113**	**0.8803**	**0.8450**	**0.8341**
**LINKNET** [58]	0.0758	0.0775	0.0781	0.8465	0.8546	0.8021	0.9601	0.9513	0.9484	0.7947	0.7720	0.7735	0.8606	0.8271	0.8965	0.7956	0.7750	0.7567
**FAPNET**	**0.0713**	**0.0732**	**0.0755**	**0.9285**	**0.8937**	**0.8544**	**0.9760**	**0.9601**	**0.9630**	**0.9371**	**0.8948**	**0.8558**	**0.9491**	**0.9133**	**0.9071**	**0.9593**	**0.8835**	**0.8363**

**Table 6 sensors-22-08245-t006:** **CFR-CB** experimental results on floodwater dataset. Average result was taken after 150 epochs with a ±2.50% error rate. **Green**, **Yellow** and **Blue** in the MeanIoU Test dataset column indicate first, second and third best, respectively.

	Focal Loss			MIoU			Dice Coff			F1			Precision			Recall		
Model	Trn	Val	Tst	Trn	Val	Tst	Trn	Val	Tst	Trn	Val	Tst	Trn	Val	Tst	Trn	Val	Tst
**UNET** [54]	**0.1009**	0.1051	0.0760	**0.9310**	0.8563	0.8456	0.8457	0.8262	0.9590	**0.7390**	0.7168	**0.8371**	**0.8319**	0.7910	0.8763	**0.7105**	0.7051	**0.8357**
**VNET** [55]	0.1032	0.1054	0.0802	0.8659	0.8463	0.7551	0.8321	0.8231	0.9343	0.6745	0.6875	0.7527	0.7848	0.8083	0.8739	0.6595	0.6773	0.7357
**DNCNN** [52]	0.1036	0.1043	0.0774	0.8567	0.8702	0.8050	0.8291	0.8226	0.9460	0.6526	0.6495	0.8010	0.7710	0.7353	0.8859	0.6456	0.6477	0.7840
**UNET++** [56]	0.1040	0.1047	0.0778	0.8302	0.8556	0.8198	0.8276	0.8219	0.9425	0.6090	0.6156	0.7888	0.7346	0.7181	0.8541	0.6109	0.6226	0.7913
**U2NET** [57]	0.1018	0.1047	0.0783	0.9014	0.8564	0.7898	0.8402	0.8253	0.9475	0.6364	0.6174	0.7611	0.7798	0.7242	0.8815	0.6337	0.6348	0.7426
**ATTUNET** [53]	0.1025	**0.1014**	0.0799	0.8392	0.8507	0.7579	**0.9601**	**0.9515**	0.9370	**0.7986**	**0.7649**	0.7592	**0.8564**	**0.8359**	0.8798	**0.8564**	**0.7650**	0.7497
**FPN** [49]	0.1033	**0.1034**	**0.0748**	0.8654	**0.8789**	**0.8651**	0.8324	0.8304	**0.9618**	0.6679	0.6839	0.8292	0.7790	0.7714	**0.9032**	0.6558	0.6797	0.8095
**LINKNET** [58]	**0.1005**	0.1066	**0.0758**	**0.9332**	**0.8959**	**0.8638**	**0.8469**	**0.9654**	**0.9650**	0.6897	**0.8598**	**0.8491**	0.8204	**0.9008**	**0.9118**	0.6730	**0.8596**	**0.8382**
**FAPNET**	**0.1002**	**0.1042**	**0.0752**	**0.9264**	**0.8797**	**0.8679**	**0.8556**	**0.8409**	**0.9668**	**0.7276**	**0.7240**	**0.8322**	**0.8417**	**0.8131**	**0.9219**	**0.6995**	**0.7116**	**0.8125**

**Table 7 sensors-22-08245-t007:** **PHR** experimental results on floodwater dataset. Average result was taken after 150 epochs with a ±2.50% error rate. **Green**, **Yellow** and **Blue** in the MeanIoU Test dataset column indicate first, second and third best, respectively.

	Focal Loss			MIoU			Dice Coff			F1			Precision			Recall		
Model	Trn	Val	Tst	Trn	Val	Tst	Trn	Val	Tst	Trn	Val	Tst	Trn	Val	Tst	Trn	Val	Tst
**UNET** [54]	0.0773	0.0772	**0.0776**	0.8241	0.8590	**0.8058**	0.9523	0.9530	0.9511	0.8119	0.6879	**0.7672**	0.8669	0.7502	**0.8452**	0.8070	**0.8021**	**0.7908**
**VNET** [55]	0.0779	0.0803	0.0800	0.8151	0.8254	0.7552	0.9501	0.9403	0.9409	0.7979	0.6730	0.6844	0.8542	0.7381	0.8377	0.7996	0.7986	0.7027
**DNCNN** [52]	0.0774	0.0778	0.0781	0.8266	0.8552	0.7959	0.9529	0.9426	0.9473	0.8090	0.6649	0.7215	0.8642	0.7028	0.8099	0.8047	0.7845	0.7478
**UNET++** [56]	0.0775	0.0769	0.0784	0.8211	0.8597	0.7878	0.9518	0.9539	0.9459	0.7977	0.6563	0.7048	0.8488	**0.7563**	0.8345	0.8010	0.7599	0.7276
**U2NET** [57]	**0.0751**	**0.0765**	0.0781	**0.8671**	**0.8719**	0.8045	**0.9652**	**0.9584**	**0.9513**	**0.8182**	0.6751	0.7350	**0.8798**	0.7479	0.8388	**0.8137**	0.7925	0.7666
**ATTUNET** [53]	0.0787	0.0779	0.0799	0.8028	0.8477	0.7496	0.9467	0.9492	0.9400	0.7857	0.6531	0.6846	0.8468	0.7181	0.8076	0.7889	0.7649	0.7084
**FPN** [49]	**0.0727**	**0.0747**	**0.0762**	**0.9065**	**0.8941**	**0.8301**	**0.9757**	**0.9656**	**0.9590**	**0.8747**	**0.6891**	**0.7660**	**0.9126**	**0.7674**	**0.8763**	**0.8628**	0.7870	**0.7743**
**LINKNET** [58]	**0.0725**	**0.0752**	**0.0755**	**0.9108**	**0.8904**	**0.8454**	**0.9766**	**0.9637**	**0.9621**	**0.8851**	**0.7101**	**0.7756**	**0.9208**	**0.7580**	**0.8706**	**0.8748**	**0.8185**	**0.7891**
**FAPNET**	0.0773	0.0772	0.0796	0.8241	0.8590	0.7649	0.9523	0.9530	0.9432	0.8119	**0.6879**	0.7216	0.8669	0.7502	0.8349	0.8070	**0.8021**	0.7498

**Table 8 sensors-22-08245-t008:** **PHR-CB** experimental results on floodwater dataset. Average result was taken after 150 epochs with a ±2.50% error rate. **Green**, **Yellow** and **Blue** in the MeanIoU Test dataset indicate first, second and third best, respectively.

	Focal Loss			MIoU			Dice Coff			F1			Precision			Recall		
Model	Trn	Val	Tst	Trn	Val	Tst	Trn	Val	Tst	Trn	Val	Tst	Trn	Val	Tst	Trn	Val	Tst
**UNET** [54]	**0.0733**	0.0821	**0.0803**	**0.9480**	0.8717	**0.7930**	**0.9720**	0.9321	**0.9429**	**0.9683**	**0.8617**	**0.7522**	**0.9686**	**0.8667**	0.7778	**0.9680**	**0.9140**	0.8311
**VNET** [55]	0.0835	0.0840	0.0826	0.8589	0.8545	0.7573	0.9191	0.9177	0.9234	0.9081	0.7998	0.7284	0.9081	0.8332	0.7780	0.9112	0.8618	0.8037
**DNCNN** [52]	0.0844	0.0802	**0.0803**	0.8496	0.8855	**0.7844**	0.9141	0.9373	**0.9380**	0.9023	0.8283	**0.7550**	0.9039	**0.8503**	**0.8128**	0.9068	0.8865	0.8146
**UNET++** [56]	0.0769	0.0802	0.0820	0.9153	0.8867	0.7664	0.9536	0.9389	0.9286	0.9460	0.7919	0.7434	0.9464	0.8275	**0.7924**	0.9464	0.8682	0.8093
**U2NET** [57]	**0.0748**	**0.0792**	0.0830	**0.9481**	0.8639	0.7615	**0.9741**	**0.9449**	0.9236	**0.9663**	0.7766	0.7423	**0.9667**	0.8093	0.7681	**0.9668**	0.8792	0.8353
**ATTUNET** [53]	0.0776	0.0800	0.0808	0.9095	0.8873	0.7776	0.9499	0.9390	0.9347	0.9406	0.7993	0.7376	0.9413	0.8262	0.7720	0.9411	0.8704	0.7993
**FPN** [49]	0.0752	0.0800	0.0901	0.9311	**0.8881**	0.6967	0.9627	0.9395	0.8900	0.9547	0.8098	0.7082	0.9546	0.8215	0.7111	0.9558	**0.8879**	**0.8497**
**LINKNET** [58]	0.0789	**0.0793**	0.0890	0.8985	**0.8941**	0.6996	0.9442	**0.9425**	0.8946	0.9342	0.8047	0.7213	0.9345	0.8176	0.7300	0.9355	0.8602	**0.8430**
**FAPNET**	**0.0730**	**0.0792**	**0.0794**	**0.9514**	**0.8960**	**0.8706**	**0.9695**	**0.9454**	**0.9400**	**0.9612**	**0.8706**	**0.7580**	**0.9565**	**0.8795**	**0.8064**	**0.9584**	**0.8967**	**0.8508**

**Table 9 sensors-22-08245-t009:** Comparison of Time and Memory Consumption during Training up to epoch 5 without compromising the accuracy. Longer training times and more computational overhead are associated with a higher refactor value. A U2NET refactoring value of 1 indicates that the model’s parameters are larger than those of competing models.

Model	Refactor	Batch	GPU (gb)	RAM (gb)	Batch Time (s)	Time for 5 Epoch (min)
**UNET** [54]	0.04	10	4.6	3.2	79	6.70
**VNET** [55]	0.33	10	7.9	3.5	95	8.08
**DNCNN** [52]	0.01	10	7.9	3.5	201	17.26
**UNET++** [56]	0.15	10	7.9	3.4	163	13.77
**U2NET** [57]	1	6	7.9	4.2	273	23.49
**ATTUNET** [53]	0.19	10	7.9	3.5	194	16.33
**FPN** [49]	0.13	10	6.6	3.4	128	10.93
**LINKNET** [58]	0.12	10	4.6	3.6	83	7.11
**FAPNET**	0.13	10	7.9	3.3	79	6.77

**Table 10 sensors-22-08245-t010:** Experimental results for individual input feature on floodwater dataset. Average result was taken after 150 epochs with a ±2.50% error rate. **Green**, **Yellow** and **Blue** indicate first, second and third best, respectively, for the test dataset in each row (MeanIoU).

Metrics	VV			VH			DEM			DFM		
	Trn	Val	Tst	Trn	Val	Tst	Trn	Val	Tst	Trn	Val	Tst
MeanIoULoss	0.8600	0.7800	**0.5405**	0.8400	0.8200	**0.6063**	0.4900	0.5200	0.5404	0.9514	0.8960	**0.8706**
	0.4800	0.3100	0.1125	0.4800	0.2900	0.1022	0.6500	0.6900	0.1280	0.0730	0.0792	0794

**Table 11 sensors-22-08245-t011:** Comparison of Time and Memory Consumption during Training up to epoch 5 without compromising the accuracy for different input features VV, VH, DEM and DFM.

Model	Batch	GPU (gb)	RAM (gb)	Batch Time (ms)	Time for 5 Epoch (min)
**VV**	10	4.6	3.3	39	3.39
**VH**	10	4.6	3.3	42	3.69
**DEM**	10	4.6	3.3	34	2.94
**DFM**	10	7.9	3.3	79	6.77

## Data Availability

Microsoft AI for Earth [8].

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
