# Peer review of "FAPNET: Feature Fusion with Adaptive Patch for Flood-Water Detection and Monitoring [Author-notes fn1-sensors-22-08245]"

_sensors, 2022, doi:10.3390/s22218245_

Round 1
Reviewer 1 Report
Overall, manuscript is very clear, and well-written. Authors intended to detect the water bodies using FAPNET model and found it working fine. Sufficient information about the previous study findings is presented for readers and is thorough. However, the quality and clarity of the images should be increased. Authors should also cross check and proof read for grammar and spelling mistakes.
Author Response
Thank you for your comments. We have improved the quality of the images and added more description for clarity. We also put more efforts on proofreading. In the new version of the document, all of the major edits are noted in red. Additional citations are included at the conclusion of this text. We observed that reviewer suggested to improve the research design and conclusion. We have attached a document for further explanation

Reviewer 2 Report
This paper used a convolutional network to detect flood water which includes numerous experiments and comparisons. However, I was not convinced by the follows:
1. The authors stated that the labels in Figure 1 contain many errors. Where are these errors (mark them on the pictures)? Where did your ground truth information come from?
2. It is heighted that a Data-Fusion Module (DFM) was proposed in this paper. However, how does it work other than stacking the three bands together?
3. I do not think your NAP augmentation module is a good choice. There are many generative augmentation methods that may do better. Samples in the test set should preferably never be presented in the training, including the transformed ones.
4. Apart from the inclusion of DFM and NAP modules and parameter settings, what are the main differences between this FAPNET model and UNET models?
5. The comparative experiments did not clearly illustrate how data used by the other models are preprocessed, and whether they are consistent with the model proposed in this paper.
I suggest the authors rewrite this article completely and resubmit it as a new paper.
Author Response
Thank you for the constructive input. We have implemented substantial changes to address the reviewers’ concerns. Modifications/revisions made in the manuscript are highlighted in red. We have attached a document for further explanation.

Reviewer 3 Report
Reviewer comments
Thank you for submitting your paper to Journal of Sensors. I read carefully manuscript number: sensors-1948362, the manuscript entitled: " FAPNET: Feature Fusion with Adaptive Patch for Floodwater Detection and Monitoring ". This paper proposes several techniques: a multi-channel Data Fusion Module (DFM), Neural Patch (NAP) augmentation algorithm, and re-weight class balancing (PHR-CB experiment), which are integrated into our novel Fusion Adaptive Patch Network (FAPNET). Techniques mentioned above used the Floodwater map from the Radar Imagery competition data hosted by Microsoft AI for Earth, which consists of three annotated classes (Water, Background, and Unlabeled) collected from 13 global flood occurrences. The authors of the paper designed four experimental setups and compared their results with the popular segmentation models UNET, VNET, DNCNN, UNET++, U2NET, ATTUNET, FPN, and LINKNET. In my point of view, the result of this kind of research could be interesting and useful for many applications specifically for the spatial inundation mapping. Please check the English grammar. The English language is moderate. Please check all parts of the manuscript and correct grammatical errors. Some sections of paper require major revisions before any further. I attached my reviewer supplementary comments in the below and pdf file.
1- Abstract
1-1- The abstract section need to complete with more information. The abstract should be improved.
1-2-The concrete finding of this research need to be added to the abstract section.
2- Introduction
2-1- In the literature review section, use newer references related to flood identification from 2020 to 2022.
2-2- The literature review is too general and thus can’t indicate any novelty of the current study. It is better that explain more about the novelty of manuscript in introduction section. The manuscript has not quite innovative. Please explain about its novelty.
3- Research Methodology section were provided in poor way. So needs improvements:
3-1- It is suggested that a section titled "Study Area(s)" be created to depict the study area(s).
3-2- Methodologies used in the manuscript should describe clearly.
3-3- What is the reason why the authors do not use optical images such as Landsat-8 and Sentinel-2 images?
3-4- The developed models by the authors can be applied to these images?
3-5- In that case, what effect will the use of a new dataset have on the number of training samples? Similarly, about the calculation time.
4 -"Results and Discussion" were provided in poor way.
4-1-Results of this study need to be compared with previous research works. Authors are emphatically recommended to provide a new section for this purpose.
4-3- Is it possible to evaluate the importance of each image or feature?
5- Conclusion section need rewriting.

Author Response
We sincerely appreciate your time and effort putting into the review. Your expertise and thoughtful comments inspire our further research in this topic. As such, we have carefully refined the manuscript to address the reviewers’ concerns, with the goal to share our scientific findings with the research community. We also play more attention on the English writing. Major changes are highlighted RED in the revised manuscript. Additional references are added at the end of this document. We have attached a document for further explanation.

Round 2
Reviewer 2 Report
I have no further comments.
Reviewer 3 Report
Thank you for submitting your revised paper to Journal of Sensors. I read the revised manuscript number: sensors-1948362, the manuscript entitled: " FAPNET: Feature Fusion with Adaptive Patch for FloodWater Detection and Monitoring". In my point of view, the result of this kind of research could be interesting and useful for many applications specifically for the spatial inundation and risk mapping of multi-index analysis. All previous comments were applied. The authors applied all comments point by point and I confirm their revision. The added information is important and useful and led to improving the manuscript. I accept the revised manuscript in this present form. I concur; the final decision is accepted for publication.